# The Central Spike Complex of Bacteriophage T4 Contacts PpiD in the Periplasm of *Escherichia coli*

**DOI:** 10.3390/v12101135

**Published:** 2020-10-06

**Authors:** Sabrina Wenzel, Mikhail M. Shneider, Petr G. Leiman, Andreas Kuhn, Dorothee Kiefer

**Affiliations:** 1Institute of Biology, University of Hohenheim, 190h, Garbenstr. 30, 70599 Stuttgart, Germany; Sabrina.Wenzel@uni-hohenheim.de (S.W.); Dorothee.Kiefer@uni-hohenheim.de (D.K.); 2Shemyakin-Ovchinnikov Institute of Bioorganic Chemistry, Laboratory of Molecular Bioengineering, 16/10 Miklukho-Maklaya St., 117997 Moscow, Russia; mm_shn@mail.ru; 3Department of BMB, Basic Sciences Building 6.600D, University of Texas Medical Branch, 301 University Blvd, Galveston, TX 77555-0647, USA; pgleiman@utmb.edu

**Keywords:** bacteriophage T4, tail structure, periplasmic chaperone, DNA translocation

## Abstract

Infecting bacteriophage T4 uses a contractile tail structure to breach the envelope of the *Escherichia coli* host cell. During contraction, the tail tube headed with the “central spike complex” is thought to mechanically puncture the outer membrane. We show here that a purified tip fragment of the central spike complex interacts with periplasmic chaperone PpiD, which is anchored to the cytoplasmic membrane. PpiD may be involved in the penetration of the inner membrane by the T4 injection machinery, resulting in a DNA-conducting channel to translocate the phage DNA into the interior of the cell. Host cells with the *ppiD* gene deleted showed partial reduction in the plating efficiency of T4, suggesting a supporting role of PpiD to improve the efficiency of the infection process.

## 1. Introduction

The adsorption and penetration of myophage T4 is a fascinating example of a self-triggered multistep event in molecular biology. The process is divided into five steps: (1) the adsorption of long tail fibres; (2) the binding of short tail fibres; (3) tail-sheath contraction; (4) tail-tube movement through the outer membrane and periplasm; and (5) the contact of the tail-tube tip with the inner membrane to initiate the phage DNA ejection process.

Bacteriophage T4 contacts its host cells by initially stochastic low-affinity binding six long tail fibres to the outer membrane [1,2,3]. This binding triggers a conformational change in the geometry of the baseplate that results in stretching out six short tail fibres (gp12), leading to an irreversible and firm binding of the phage to the bacterial lipopolysaccharide moiety of the cell [4,5]. The conformational transition of the baseplate to a starlike configuration then starts the contraction of the tail structure [6]. Thereby, the gp18 proteins of the tail sheath rearrange their intersubunit contacts, leading to wavelike shortening of the sheath [7,8]. Consequently, the tail sheath is shortened from 92.5 to 42 nm, whereas the inner tail tube stays in its stiff cylinder-like structure that is mechanically forced to move through the centre of the baseplate.

The movement of the tail tube towards the cell surface takes baseplate components gp5.4, gp5, gp27, gp48, and gp54 along at the leading end of the tube (gp19); gp5.4, at the tip of this structure, is tightly connected to the C-terminal part of gp5, which forms a stable triple helix, and is called the gp5C-gp5.4 central spike complex (Figure 1A) [9,10].

Gp5 exists in the phage as a cleaved protein, with the N-terminal part (gp5*) bound to gp27 and the C-terminal part tightly bound to gp5.4. The penetration of the outer membrane is most likely executed by mechanical force generated by the sheath contraction and applied to the central spike structure. In the periplasm, the tip of the tail moves through the peptidoglycan layer, which is facilitated by the mureolytic activity of the lysozyme domain of gp5.

The molecular details that lead to contact with the inner membrane and to the translocation of the phage DNA through the membrane into the cytoplasm are still unknown. Here, a number of scenarios are discussed. The needle at the tip of the tail tube may come into contact with the inner membrane and initiate an opening of a DNA-conductive channel. Second, the central spike complex dissociates from the tail tube in the periplasm, leaving the N-terminal part of gp5 exposed on the tube tip, which could then come into contact with the inner membrane. Furthermore, it is possible that, in the periplasm, the entire T4 tip structure dissociates from the tube, and gp27 or gp48 comes into contact with the inner membrane for DNA translocation.

To shed some light on these interesting events, we purified the gp5Cβ–gp5.4 complex, a tip fragment of the central spike complex, and tested its binding to liposomes and spheroplasts. We found that this spike fragment binds to periplasmic protein PpiD that is anchored to the inner membrane. PpiD is a periplasmic chaperone [12] that is connected to the SecYEG translocon [13]. A chromosomal deletion of PpiD resulted in a decrease of T4 plating efficiency on this strain, suggesting that PpiD might play an important role in T4 infection. 

## 2. Materials and Methods

### 2.1. Bacterial Strains and Phage

T4 was propagated on *E. coli* B and on BW25113 (parental strain of *ΔppiD*; Keio collection) in Luria broth (LB medium). Media preparation and bacterial growth were performed according to standard methods [14]. *ppiD* deletion strain JW0431 was cultured in LB medium containing 25 µg/mL kanamycin at 37 °C. *E. coli* BL21 (DE3) [15], transformed with plasmid pEEva2 encoding the gp5Cβ–5.4 complex or plasmid pASK-IBA3C encoding PpiD, respectively, was grown in LB in the presence of ampicillin (100 µg/mL, final concentration).

### 2.2. Purification of gp5Cβ–gp5.4 Complex

*E. coli* BL21 (DE) harbouring plasmid pEEva2 encoding the His–gp5Cβ–gp5.4 complex was grown in a 2xYT medium containing 100 µg/mL ampicillin at 37 °C until the culture had reached OD_600_ of 0.6. The culture was cooled down to 18 °C. Gp5Cβ–5.4 complex expression was induced by the addition of isopropyl-β-D-thiogalactopyranoside (IPTG) to 0.5 mM, and the culture was grown at 18 °C overnight. Cells were harvested by centrifugation at 8000× *g* at 4 °C for 15 min. Cell pellets were resuspended in a lysis buffer (20 mM Tris-HCl (pH 8), 300 mM NaCl, 5 mM imidazole) on ice and lysed using the OneShot (Constant Systems) at 1.23 kbar. Before cell disruption, 1 mM phenylmethylsulfanyl-fluoride (PMSF) was added. The lysate was then centrifuged at 35,000× *g* and 4 °C for 20 min. The supernatant was loaded into a 2 mL Ni-Sepharose 6 Fast Flow matrix. After consecutively washing the column with 10 mL of each of the two wash buffers (20 mM Tris-HCl (pH 8), 300 mM NaCl, 37.5 mM imidazole (Wash Buffer 1) or 50 mM imidazole (Wash Buffer 2)), the complex was eluted with 20 mL elution buffer (20 mM Tris-HCl (pH 8), 150 mM NaCl, 300 mM imidazole) in 2 mL fractions. The elution fractions containing the target protein were further purified by ion-exchange chromatography performed with a Resource Q 1 mL column connected to an ÄKTA purifier 10 system. The sample was loaded onto the column that was equilibrated with Buffer A (20 mM Tris-HCl (pH 8)) and eluted with a linear gradient against Buffer B (20 mM Tris-HCl (pH 8), 1 M NaCl). Relevant fractions were combined and concentrated using Sartorius Vivaspin centrifugal concentrators with a molecular weight cut-off of 10 kDa. This sample was loaded onto a Superdex 75 10/300 GL column connected to an ÄKTA purifier 10 system, and purified in a gel filtration buffer (10 mM Tris-HCl (pH 8), 150 mM NaCl).

### 2.3. Size-Exclusion Chromatography–Multiple-Angle Light Scattering (SEC-MALS)

Protein samples were injected into a Superdex 200 increase 10/300 GL column, equilibrated with a gel filtration buffer (10 mM Tris-HCl (pH 8), 150 mM NaCl). The SEC column was linked to a static 3-angle light-scattering detector (miniDAWN Treos II) and a refractive-index detector (Optilab T-rEX) (Wyatt Technology, Santa Barbara, CA, USA). Data were collected every second at a flow rate of 0.5 mL/min. Data analysis was carried out using ASTRA VII software, yielding the molar mass for each fraction. In addition, protein-conjugate analysis was performed to calculate the molar mass of each component of a protein complex. The light-scattering detectors were normalised, and data quality was assessed by a BSA standard (Thermo, Waltham, MA, USA).

### 2.4. PpiD Purification 

*E. coli* PpiD was expressed from plasmid pASK-IBA3C encoding PipD-Strep (kindly provided by HG Koch, Freiburg) in BL21 (DE3) cells in LB medium containing 25 µg/mL chloramphenicol at 37 °C until the culture had reached an OD_600_ of 0.6. The culture was induced by the addition of anhydrotetracycline to a final concentration of 200 µg/L for 2 h at 37 °C. Cells were harvested by centrifugation at 10,000× *g* and 4 °C for 15 min. Cell pellets were resuspended in Buffer A (50 mM Tris-HCl (pH 7.4), 50 mM NaCl, 1 mM Ethylen-diamino-tetra-acidic acid (EDTA)) on ice and lysed using the OneShot instrument at 1.23 kbar. Before cell disruption, 1 mM PMSF was added. The lysate was then centrifuged at 35,000× *g* and 4 °C for 45 min. The pellet was resuspended in Buffer A containing 1% N-lauroylsarcosine to solubilise the inner membrane at 4 °C overnight. The solubilised membrane fraction was collected by ultracentrifugation at 110,000× *g* at 4 °C for 40 min and loaded onto 1 mL of Strep-Tactin XT matrix. After washing the matrix with 5 mL of Buffer W (100 mM Tris-HCl (pH 8), 150 mM NaCl, 1 mM EDTA), the protein was eluted with 3 mL of Buffer BXT (100 mM Tris-HCl (pH 8), 150 mM NaCl, 1 mM EDTA, 50 mM biotin) in 500 µL fractions. The elution fractions containing the target protein were further purified by gel filtration chromatography performed with a Superdex 200 increase 10/300 GL column connected to an ÄKTA purifier 10 system in Buffer A.

### 2.5. Spheroplast Preparation 

For cross-linking and binding assays, spheroplasts were generated after centrifugation of *E. coli* BL21 (DE3) harbouring plasmid pMAL-p5X (NEB) at 5000× g and 4 °C for 15 min. The cell pellet was resuspended in an ice-cold spheroplast buffer (20 mM Tris-HCl (pH 8), 0.5 M sucrose) and lysozyme (20 µg/mL), and incubated for 5 min at room temperature (RT). Then, 1.6 mM EDTA was added, and the sample was incubated for 7 min at RT [16]. Finally, 10 mM MgCl_2_ was added to the spheroplasts for stabilization.

### 2.6. Liposomes and Proteoliposomes

*E. coli* lipids (Avanti) 1-palmitoyl-2-oleoyl-*sn*-glycero-3-phospho-ethanolamine (POPE) and 1-palmitoyl-2-oleoyl-*sn*-glycero-3-phosphoglycerol (POPG) were dissolved in dichloromethane and evaporated under a stream of nitrogen. The dry lipid film was dissolved in Buffer A (50 mM Tris-HCl (pH 7.4), 50 mM NaCl, 1 mM EDTA). A 7:3 mixture of PE and PG was used, mimicking the membrane-lipid composition of *E. coli*. Liposomes with an average diameter of 250 nm were generated by extrusion (Mini extruder, Avanti Polar Lipids. Inc.) 31 times through a 0.4 µm polycarbonate membrane (Whatman). For preparing the proteoliposomes, purified PpiD protein was mixed with a lipid/protein ratio of at least 10,000:1 (mol/mol) and extruded as described above.

## 3. Results

### 3.1. Expression and Purification of Central Spike Complex

The T4 central spike complex was purified from *E. coli* BL21 harbouring plasmid pEEva2, encoding a C-terminal fragment of the gp5C β-helix comprising amino acid residues 484–575 and gp5.4 (Figure 1 and Figure 2A). Gp5Cβ contains an N-terminal 6xHis-tag followed by a TEV cleavage site. Full-length C-terminal β helix (352-575) is a natural cleavage product of gp5 that appears during assembly of the T4 phage particle [17]. The central spike complex was expressed from cells that were grown overnight at 18 °C in a shaking incubator. The protein complex was affinity-purified by Ni–NTA, followed by ion-exchange chromatography and gel filtration, resulting in a fraction that mainly consisted of the trimeric complex of gp5Cβ with a bound gp5.4 monomer giving an apparent molecular weight (MW) of the complex of about 38 kDa in SDS-PAGE (Figure 2B). Most of the complex was therefore stable in SDS, and only a fraction was dissociated into separate components: a trimeric gp5Cβ (3 × 11.9 kDa) and a gp5.4 monomer of about 11 kDa with apparent MWs of 26 and 12 kDa, respectively.

The stoichiometry of the complex was analysed by size-exclusion chromatography and multiangle light scattering (SEC-MALS). The purified gp5Cβ–5.4 complex was eluted in a single peak with a retention time of 29.16 min (Figure 3A). Total molecular weight was determined to be 45.4 kDa (Figure 3B).

In addition, protein-conjugate analysis was performed to calculate the molar mass of each component of the conjugate complex. The molar mass of gp5Cβ and gp5.4 was determined to be 34.5 and 10.8 kDa, respectively. These fit with the calculated MW of the complex of 46 kDa, consisting of a triple gp5C β helix and one copy of gp5.4. Following the molecular weight across the entire peak showed homogeneous distribution both of the complex and of each component. Treatment of the complex with 15% tri-chloro-acetic acid (TCA) resulted in the dissociation of gp5.4 from gp5Cβ, but leaving the triple helix intact. SDS gel electrophoresis revealed two bands with apparent molecular weights of about 26 and 12 kDa, respectively (Figure 2B). Taken together, MALS analysis corroborated an exact 3:1 molecular ratio between gp5Cβ–helix and gp5.4. It also demonstrated that the complex was homogeneous and stable in solution.

### 3.2. The gp5Cß–gp5.4 Complex Does Not Bind to Liposomes and Weakly to Spheroplasts

Since it is plausible that the central spike complex is involved in contacting or perforating the inner membrane, the purified gp5Cβ–gp5.4 complex was tested for its binding to liposomes and spheroplasts. The liposomes were generated from 70% 1-palmitoyl-2-oleoyl-phospho-ethanolamine (POPE) and 30% 1-palmitoyl-2-oleoyl-phospho-glycerol (POPG) with an extruder to obtain an average diameter of 250 nm of unilamellar liposomes. The purified gp5Cβ–gp5.4 complex was added to the liposomes at room temperature for 30 min and then applied to an airfuge at 20 psi for 10 min. The supernatant and liposomes in the pellet were then analysed by SDS-PAGE (Figure 4A). The gp5Cβ–gp5.4 complex and gp27 control remained in the supernatant, whereas the SecA protein used as an additional control showed binding to the liposomes.

Spheroplasts were prepared from *E. coli* BL21 by osmotic-shock treatment with lysozyme and EDTA for 10 min. 20 mM MgCl_2_ was added to the cell suspension to stabilise the spheroplasts. Then, 25 µg of the purified gp5Cβ–5.4 complex was added to the spheroplasts and incubated at room temperature for 20 min, followed by centrifugation at 5000× *g*. The supernatant and spheroplast pellet were analysed by cross-linking and Western blotting (Figure 4B). Only small amounts of the purified complex were found in association with the spheroplasts.

### 3.3. Interaction of gp5Cß–gp5.4 Complex with PpiD in the Periplasm

Most likely, the central spike complex remains in the periplasm after penetration through the outer membrane [2]. To investigate whether it transiently interacts with a periplasmic component on the periplasmic surface of the inner membrane, spheroplasts were prepared, followed by cross-linking treatment (Figure 5). The purified gp5Cβ–gp5.4 complex (Lane 2) was incubated with the spheroplasts for 10 min in the presence of 0.5% formaldehyde at room temperature, purified by affinity chromatography, and analysed by SDS-PAGE (Lanes 1 and 3). For the control, an aliquot of the formaldehyde-treated sample was heated to 100 °C for 10 min (Lane 3), which resulted in a separation of the cross-links. The formaldehyde treatment resulted in an additional band at about 75 kDa (indicated by an asterisk in Lane 1) that was further processed for matrix-assisted laser-desorption/-ionization (MALDI) analysis [18]. This analysis revealed gp5C and the PpiD protein as cross-link products with a probability of over 95%.

To verify such an interaction, PpiD was purified and added to the liposomes in a reconstitution setup. The resulting proteoliposomes containing PpiD were then tested for binding the gp5Cβ–gp5.4 complex (Figure 6A). The PpiD proteoliposomes were incubated with the purified complex at room temperature for 10 min and sedimented by centrifugation. The proteoliposomes were indeed capable to reproducibly bind a small portion of the complex (Lane 3), in contrast to empty liposomes (Lane 5). This result is consistent with the observation of the cross-linking experiment, and suggests that PpiD might play a supporting role for the infection process of T4. To test this idea, the plating efficiency of T4 was analysed with the *ppiD* deletion strain JW0431Δ*ppiD* and compared to the parental BW25113 (Figure 6B). The plating efficiency of T4 on JW0431Δ*ppiD* was reduced to 78% ± 2% (Figure 6B). When the plasmid-encoded PpiD was expressed in JW0431Δ*ppiD*, plating efficiency was restored.

### 3.4. The Fate of the Central Spike Complex after the Infection Process

Several gp5 and gp5.4 peptide-related antisera were raised and tested on purified phage preparations. A serum recognising gp5C in the phage was directed against the C-terminal 20 residues (537–556) of gp5C with antigenic peptide KVAGTVDWDVGGDWTEKMAS. This serum allowed for recognising the gp5C-containing central spike complex in the phage particle (Figure 7A), and to follow the fate of the protein after infection of *E. coli* BL21 bearing plasmid pMAL-p5X, expressing MalE (Figure 7B,C). Exponentially growing cells were infected with a multiplicity of infection (MOI) of 20, and samples were taken at 2, 5, and 10 min postinfection (pI). Samples were spun down to remove the nonadsorbed phage, and infected cells present in the pellet were converted to spheroplasts. The released periplasmic shock fluids (odd-numbered lanes) were separated from the spheroplasts (even-numbered lanes). Samples were separated by PAGE and analysed by Western blot. Results showed that the gp5C-containing complex was present in the pellet fraction (Lanes 4, 6, and 8), and also progressively appeared in the periplasmic fluid (Lanes 3, 5, and 7). The complex remained stable at least until 10 min pI (Lanes 7 and 8).

## 4. Discussion

The infection process of myophage T4 involves an astounding cascade of conformational events, and this is exemplary in protein biochemistry. Remarkably, the self-triggered multistep process is independent of any outside energy input, similar to the mechanism of an old-fashioned time watch driven to operate for days by a key-loaded spring. The first steps of the phage adsorption and interaction of tail fibres with the receptors of the host cell [3], the movement of the short fibres, and the contraction of the tail sheath [4,5] are quite well-understood, whereas the molecular events that occur when the phage is penetrating the outer and inner host membrane are still unknown.

When the tail tube, driven by sheath contraction, moves through the baseplate, it takes with itself the central hub proteins of the baseplate, gp5, gp5.4, and gp27, together with gp48 and gp54. Therefore, the structure that enters the host periplasm is a capped tail tube structure, as depicted in Figure 1. In the present study, we followed the fate of the gp5 and gp5.4 proteins at the leading end of the periplasm-entering tail tube. In the phage particle, gp5 is a cleaved protein that is present as two distinct fragments, gp5* and gp5C [17]. Gp5C is folded into a triple helix that tightly binds 5.4 and withholds its triple-helical structure [9], even after denaturing treatments on SDS-PAGE giving rise to an apparent 48 kDa protein band (Figure 7A). The stability of the complex allowed for purifying a fragment of gp5C (484–575), termed gp5Cβ, with gp5.4 from *E. coli* cells encoded on a plasmid [10]. Expressed from the *E. coli* cells, the two proteins formed a complex, gp5Cβ–gp5.4, which was partially stable on SDS-PAGE (Figure 2B), and encompassed 3 copies of gp5Cβ and one copy of gp5.4, as was verified by MALS analysis (Figure 3). 

The purified complex was then used for binding experiments in vitro to uncover possible interacting partner proteins. First, the addition to spheroplasts and liposomes did not show strong interaction (Figure 4), suggesting that the central spike complex freely diffuses in the periplasm, as was suggested earlier [19]. However, when we used formaldehyde as a chemical cross-linker to stabilise possible interactions with spheroplasts, we found that PpiD copurified with the gp5Cβ–5.4 complex (Figure 5). This interaction was verified with proteoliposomes that were reconstituted with purified PpiD (Figure 6A). PpiD proteoliposomes showed enhanced binding of the added central spike complex when compared with that of pure liposomes. PpiD is a periplasmic chaperone, anchored to the inner membrane that is devoted to binding and stabilising proteins. Regarding its function for T4 infection, it could either catalyse the interaction of the tail tube with the inner membrane for penetration or simply bind the already dissociated gp5Cβ–gp5.4 complex to prevent an inhibitory role for efficient tail tube–inner membrane interaction. Additional experiments are required to clarify these points.

The involvement of PpiD in the T4 infection and propagation process was then studied with efficiency of plating (EOP). This assay quantified the proportion of infecting phage that successfully led to the formation of a plaque. When we compared the parental strain with the deletion mutant of PpiD, a reduction in EOP to less than 80% was determined. This points to a nonessential but supporting role of PpiD in the T4 infection process. Alternatively, other periplasmic chaperones such as SurA or Skp [20] might step in when *ppiD* is deleted and compensate for the loss of PpiD.

When we followed the gp5C-containing complex in phage particles, the major portion was found in a 48 kDa protein band (Figure 7A,B). Minor bands migrated at 65 and 52 kDa, respectively. The known cleavage site at amino acid residue 351 generates a triple helix of gp5C with a theoretical molecular weight of 59 kDa. Previously, it was shown that gp5C runs with a lower apparent MW [11], which might be caused by protein folding of the triple helix. After infection, the 48 kDa protein complex remained mainly with the spheroplasts (Figure 7B). Therefore, the central spike complex might indeed come into contact with the cytoplasmic membrane. However, in our analysis with the purified gp5Cβ–gp5.4 complex, we did not find a strong binding to spheroplasts. It is, therefore, possible that additional factors are required to support the binding of the T4 tail tube to the membrane surface.

Taken together, our results suggest that the T4 central spike complex enters the periplasm and interacts with low affinity to PpiD. In our analysis, we did not observe stable interaction with the inner membrane, which is required for the phage to establish a DNA-translocating complex.

## Figures and Tables

**Figure 1 viruses-12-01135-f001:**
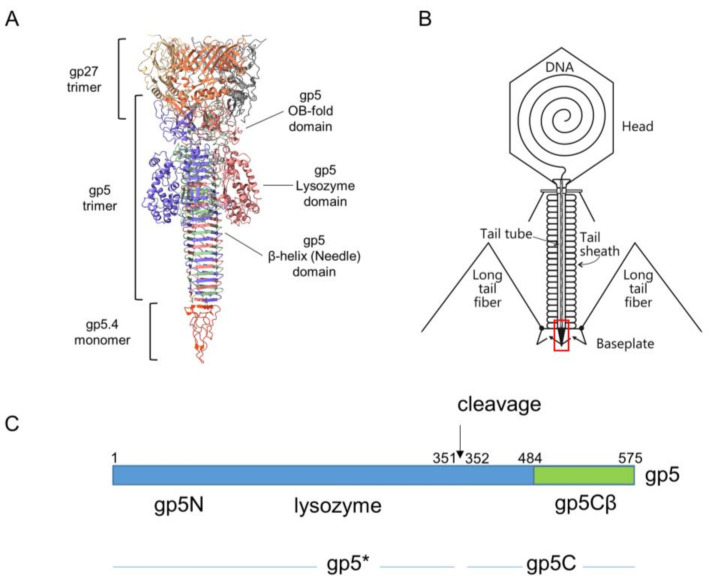
(**A**) Ribbon diagram of T4 central spike complex extracted from T4 baseplate structure [5]. Each chain shown in distinct colour. The gp27 hub protein forms an extension to the tail tube. It holds the N-terminal oligosaccharide/oligonucleotide binding fold (OB-fold) domain of the gp5 spike protein in its central channel. The gp5 spike also contains a middle lysozyme domain and a C-terminal β helix domain. The blunt tip of the β helix is sharpened by spike tip gp5.4. (**B**) Schematic structure of the T4 phage particle [8]. Location of the spike complex shown as a red rectangle. (**C**) Schematic of the gene *5* derived products. The gp5 protein is cleaved at position 351 after its assembly into the phage, resulting in gp5* and gp5C [11]. The carboxyl-terminal part of gp5C (green) encompassing the triple β helix that starts at residue 484 was cloned onto a plasmid [10].

**Figure 2 viruses-12-01135-f002:**
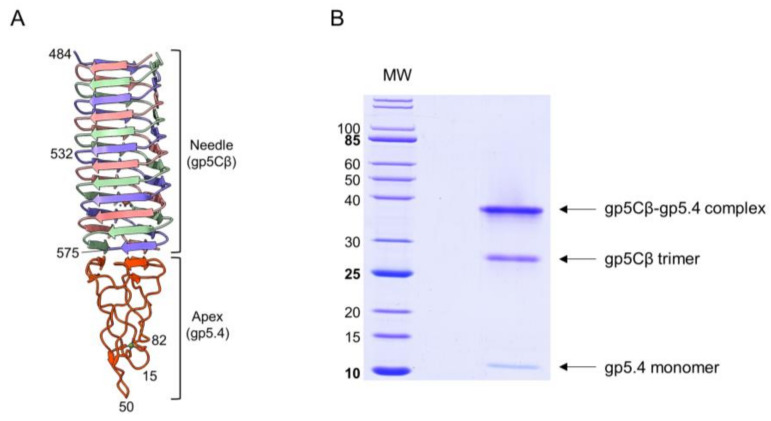
(**A**) Fragment of central-spike-complex structure comprising the amino acid residues 484–575 of gene 5 with the entire gp5.4. Each chain is a distinct colour. Residue numbers given for strategic positions. (**B**) Purification of central spike complex. Coexpression of the amino acid residues 484–575 of gp5 and gp5.4 resulted in the gp5Cβ–gp5.4 complex (apparent MW = 38 kDa). The complex partially dissociated after SDS-PAGE into a 26 kDa gp5Cβ trimeric helix and a 12 kDa monomeric 5.4 apex.

**Figure 3 viruses-12-01135-f003:**
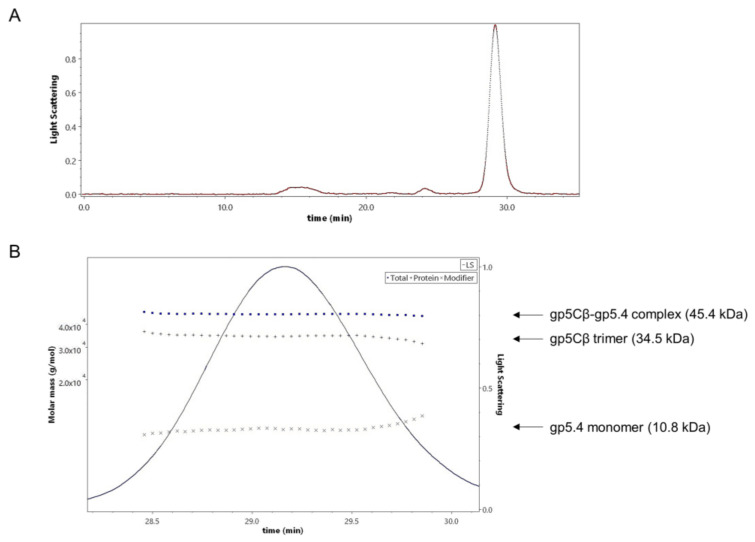
(**A**) Size-exclusion multiangle-light-scattering (SEC-MALS) analysis of gp5Cβ–gp5.4 complex. Affinity-purified complex separated on a Superdex 200 increase where it was eluted at a retention time of 29.16 min. (**B**) Fractions containing the complex were analysed with a 90° light-scattering detector verifying its composition and the stoichiometry of its components. The complex was composed of 3 molecules of the β helix and one molecule of the gp5.4 protein. Calculated molecular weights (dotted lines, left *Y* axis) were consistent with theoretical molecular weights.

**Figure 4 viruses-12-01135-f004:**
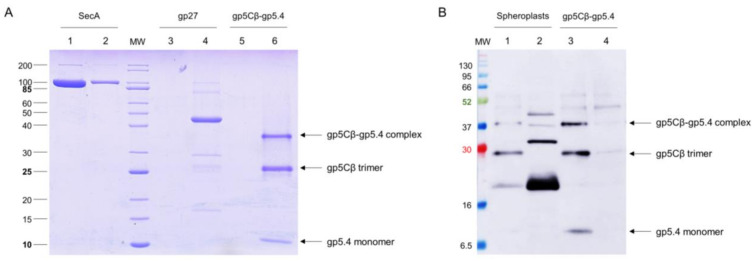
(**A**) The gp5Cβ–gp5.4 complex does not bind to liposomes. The purified complex was incubated at room temperature for 30 min to preformed liposomes (70% 1-palmitoyl-2-oleoyl-phospho-ethanolamine (POPE), 30% 1-palmitoyl-2-oleoyl-phospho-glycerol (POPG)) and separated after centrifugation (Lane 5 pellet, Lane 6 supernatant). As controls, purified SecA protein (Lane 1 pellet, Lane 2 supernatant) and purified T4 gp27 (Lane 3 pellet, Lane 4 supernatant) were added to the liposomes and centrifuged. SecA was found in the pellet (Lane 1), and the gp5Cβ–gp5.4 complex (Lane 6) and gp27 (Lane 4) were found in the supernatant. (**B**) The gp5Cβ–gp5.4 complex binds weakly to spheroplasts. Incubation of the complex for 20 min with spheroplasts was followed by centrifugation, SDS PAGE, and a Western blot with antiserum to gp5.4 (Lanes 1 and 2). A portion of the complex was found in the pellet fraction (Lane 2), whereas the control without spheroplasts mostly showed the gp5Cβ–gp5.4 complex in the supernatant (Lane 3, Lane 4 pellet).

**Figure 5 viruses-12-01135-f005:**
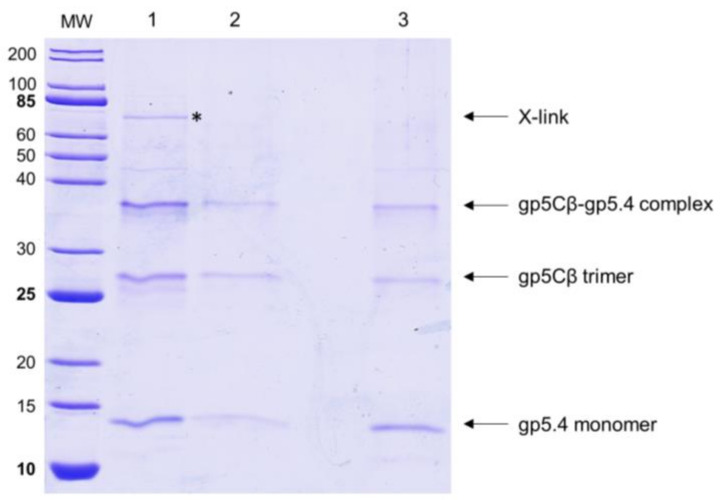
The gp5Cβ–gp5.4 complex interacts with spheroplast PpiD. The purified complex was incubated with spheroplasts in the presence of 0.5% formaldehyde for 10 min (Lanes 1 and 3). After affinity purification, the complex coeluted with periplasmic protein PpiD, which was identified by matrix-assisted laser-desorption/-ionization (MALDI) analysis (Lane 1, asterisk). As control, the affinity-purified gp5Cβ–gp5.4 complex is shown (Lane 2). To break the cross-link, sample was heated at 100 °C (lane 3).

**Figure 6 viruses-12-01135-f006:**
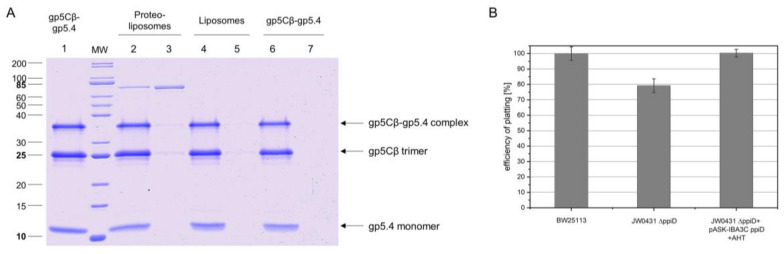
(**A**) Binding of gp5Cβ–gp5.4 complex to PpiD proteoliposomes. Purified PpiD protein was reconstituted into proteoliposomes (70% POPE, 30% POPG). Purified complex (Lane 1) was added and incubated at room temperature (RT) for 10 min. After spinning down proteoliposomes, the complex copelleted with the PpiD proteoliposomes (Lane 2 supernatant, Lane 3 pellet), but it did not bind to liposomes (Lane 4 supernatant, Lane 5 pellet) nor was it present as aggregate (Lane 6 supernatant, Lane 7 pellet). (**B**) Efficiency of plating (EOP) is affected by deletion of the *ppiD* gene. JW0431 cells, deficient for expressing PpiD and its parent strain BW25113, were tested for their efficiency to form plaques after T4 infection. Number of plaques was reduced to 78% ± 2%, an indication that T4 phage infection is less efficient when PpiD is missing. Plasmid-encoded PpiD in the JW0431 deletion strain restored EOP to 100%.

**Figure 7 viruses-12-01135-f007:**
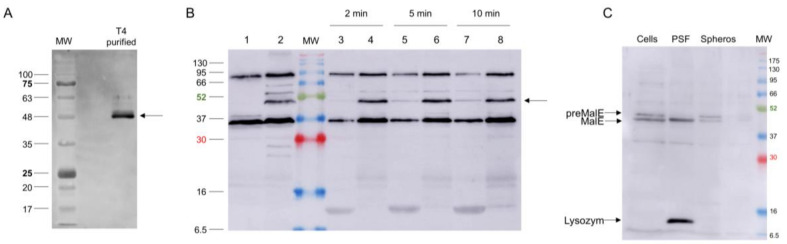
(**A**) Presence of gp5C in CsCl-purified T4 phage visualised by Western blot. (**B**) Fate of gp5C (bands marked with an arrow) from infecting phage at 2 (Lanes 3 and 4), 5 (Lanes 5 and 6) and 10 min post infection (pI; Lanes 7 and 8). Gp5C found in spheroplasts (Lanes 4, 6, and 8) and periplasmic fluid (Lanes 3, 5, and 7). For control, uninfected (Lane 1) and whole T4-infected cells (10 min pI; Lane 2) were analysed. (**C**) Fractionation of pre-MalE and MalE in periplasmic shock fluid (PSF) and in spheroplasts (Spheros) as controls.

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
