# Peer review of "The Central Spike Complex of Bacteriophage T4 Contacts PpiD in the Periplasm of Escherichia coli"

_viruses, 2020, doi:10.3390/v12101135_

Round 1
Reviewer 1 Report
939271
The manuscript suports contact of the central spike complex of bacteriophage T4 with host periplasmic chaperone PpiD in the host periplasm to initiate infection by transfer of the DNA.
The biochemistry and structural biology is excellent, convincing, and fully supports the conclusion that the spike tip contact with host cell protein PpiD is important for initiating infection.
Nevertheless it is surprising that this interaction is of such minor consequence for initiation of infection: plating efficiency is reduced only~20% in the absence of PpiD. This is not a "l. 69 marked decrease". The authors should offer some explanation or hypothesis for how infection proceeds in the absence of the PpiD protein. Is there some comparable backup chaperone for an alternate delivery mechanism?
Very minor errors are noted here:l.65 T4 tip, l. 145, that were grown, l. 159, determined as 45.4 kDa, l. 191, were treated accordingly ?, l. 213, a general reference to MALDI analysis should be given, l. 302, However, in our analysis...
Author Response
Reviewer #1
The manuscript suports contact of the central spike complex of bacteriophage T4 with host periplasmic chaperone PpiD in the host periplasm to initiate infection by transfer of the DNA.
The biochemistry and structural biology is excellent, convincing, and fully supports the conclusion that the spike tip contact with host cell protein PpiD is important for initiating infection.
Nevertheless it is surprising that this interaction is of such minor consequence for initiation of infection: plating efficiency is reduced only~20% in the absence of PpiD. This is not a "l. 69 marked decrease". The authors should offer some explanation or hypothesis for how infection proceeds in the absence of the PpiD protein. Is there some comparable backup chaperone for an alternate delivery mechanism?
Answer: We skipped the “marked” in this sentence. In the discussion we mentioned other possible backup chaperones present in the periplasm and added a citation for this (Mas et al.)
Very minor errors are noted here:l.65 T4 tip, l. 145, that were grown, l. 159, determined as 45.4 kDa, l. 191, were treated accordingly ?, l. 213, a general reference to MALDI analysis should be given, l. 302, However, in our analysis...
Answer: We have corrected all these points and added a reference for the MALDI-TOF (Webster and Oxley, 2011).

Reviewer 2 Report
The virus bacteriophage T4 ruptures the membrane of E. coli using a tail tube equipped with a needle-like tip, called “Central Spike Complex (CSC)”. The molecular details of the interaction between the tail tube and membrane during the injection process is still unknown. To help to understand the tail tube-membrane interaction, this paper tests the binding of CSC to liposomes and spheroplasts as well as the influence of PpiD deletion on the T4 plating efficiency. This paper can be published after addressing the following concerns:
- Although the results show that CSC can bind to PpiD in the designed experiments, it does not provide strong proof for the CSC interaction with PpiD within the cell membrane during the real injection process. The CSC might dissociate from the tail tube during the injection process before interaction with PpiD. Particularly that the deletion of ppid reduces the T4 plating efficiency only by %20 which means PpiD may not have a direct and significant role in infection. I believe that more experiments and direct observations are required to make a certain conclusion. Given this, if the authors still conclude that the CSC definitely reaches PpiD and interacts with it in the real injection process, they should provide a more supportive argument on this in the Results and Discussion. Otherwise, I suggest revising all explicit claims within the paper about the interaction of CSC with PpiD including the Title, Abstract, and body.
- The authors studied the effect of ppid deletion on the T4 plating efficiency. What do they expect in the case of overexpression of ppid? If possible, the authors are recommended to add experiments or discussion about the effect of overexpression of ppid on the infection efficiency.
- In the Method section “Liposomes with a diameter of 250 nm were generated….”, and in the Results section ”… to obtain an average diameter of 400 nm of unilamellar liposomes”. Please check if the numbers are consistent.
- A major part of the Results section has been devoted to the experimental details rather than new results and arguments. The authors should provide more informative arguments and discussions if applicable.
- All figures should be rendered with higher quality.
Author Response
Reviewer #2
The virus bacteriophage T4 ruptures the membrane of E. coli using a tail tube equipped with a needle-like tip, called “Central Spike Complex (CSC)”. The molecular details of the interaction between the tail tube and membrane during the injection process is still unknown. To help to understand the tail tube-membrane interaction, this paper tests the binding of CSC to liposomes and spheroplasts as well as the influence of PpiD deletion on the T4 plating efficiency. This paper can be published after addressing the following concerns:
- Although the results show that CSC can bind to PpiD in the designed experiments, it does not provide strong proof for the CSC interaction with PpiD within the cell membrane during the real injection process. The CSC might dissociate from the tail tube during the injection process before interaction with PpiD. Particularly that the deletion of ppid reduces the T4 plating efficiency only by 20% which means PpiD may not have a direct and significant role in infection. I believe that more experiments and direct observations are required to make a certain conclusion. Given this, if the authors still conclude that the CSC definitely reaches PpiD and interacts with it in the real injection process, they should provide a more supportive argument on this in the Results and Discussion. Otherwise, I suggest revising all explicit claims within the paper about the interaction of CSC with PpiD including the Title, Abstract, and body.
Answer: We fully agree with the reviewer and accordingly changed the title, abstract and body to point out that purified CSC binds to PpiD. The role of PpiD in the in vivo penetration process is presently only suggestive.
- The authors studied the effect of ppid deletion on the T4 plating efficiency. What do they expect in the case of overexpression of ppid? If possible, the authors are recommended to add experiments or discussion about the effect of overexpression of ppid on the infection efficiency.
Answer: We made this experiment and included the result in figure 6B. Overproduction in the deletion strain restored the reduced EOP in the deletion strain.
- In the Method section “Liposomes with a diameter of 250 nm were generated….”, and in the Results section”… to obtain an average diameter of 400 nm of unilamellar liposomes”. Please check if the numbers are consistent.
Answer: We have corrected this. The pore size of the extruder is 400 nm leading to liposomes with an average of 250 nm diameter as determined by dynamic light scattering.
- A major part of the Results section has been devoted to the experimental details rather than new results and arguments. The authors should provide more informative arguments and discussions if applicable.
Answer: We have added some sentences in the results part (lines 183-4, 239-240) and also extended the discussion (lines 300-3, 311-4).
- All figures should be rendered with higher quality.
Answer: We have done this and exchanged figures 1 and 2. We also separately uploaded the figures with a higher resolution.
